# Introgression and disruption of migration routes have shaped the genetic integrity of wildebeest populations

Xiaodong Liu [1,10], Long Lin [1,10], Mikkel-Holger S. Sinding [1,10], Laura D. Bertola [1], Kristian Hanghøj [1], Liam Quinn [1], Genís Garcia-Erill [1], Malthe Sebro Rasmussen [1], Mikkel Schubert [2], Patrícia Pečnerová [1], Renzo F. Balboa [1], Zilong Li[1], Michael P. Heaton [3], Timothy P. L. Smith [3], Rui Resende Pinto[4,5], Xi Wang[1], Josiah Kuja[1], Anna Brüniche-Olsen [1], Jonas Meisner[2,6], Cindy G. Santander [1], Joseph O. Ogutu [7], Charles Masembe [8], Rute R. da Fonseca[4,5], Vincent Muwanika [9], Hans R. Siegismund[1], Anders Albrechtsen [1,11] ✉, Ida Moltke [1,11] ✉ & Rasmus Heller [1,11] ✉

The blue wildebeest (*Connochaetes taurinus*) is a keystone species in savanna ecosystems from southern to eastern Africa, and is well known for its spectacular migrations and locally extreme abundance. In contrast, the black wildebeest (*C. gnou*) is endemic to southern Africa, barely escaped extinction in the 1900s and is feared to be in danger of genetic swamping from the blue wildebeest. Despite the ecological importance of the wildebeest, there is a lack of understanding of how its unique migratory ecology has affected its gene flow, genetic structure and phylogeography. Here, we analyze whole genomes from 121 blue and 22 black wildebeest across the genus' range. We find discrete genetic structure consistent with the morphologically defined subspecies. Unexpectedly, our analyses reveal no signs of recent interspecific admixture, but rather a late Pleistocene introgression of black wildebeest into the southern blue wildebeest populations. Finally, we find that migratory blue wildebeest populations exhibit a combination of long-range panmixia, higher genetic diversity and lower inbreeding levels compared to neighboring populations whose migration has recently been disrupted. These findings provide crucial insights into the evolutionary history of the wildebeest, and tangible genetic evidence for the negative effects of anthropogenic activities on highly migratory ungulates.

The blue wildebeest (*Connochaetes taurinus*) is one of the most iconic ungulates in the world, well-known for its epic annual migrations in several parts of its range. For instance, nearly 1.4 million individuals cross the plains of the Serengeti-Mara[1], making it the largest ungulate migration[2] and one of the most recognizable wildlife spectacles in the world. Their abundance and migratory behavior make them a key species for the vegetation turnover in many ecosystems, particularly in the archetypical acacia savannas of eastern and southern Africa[3,4].

The blue wildebeest is divided into five geographically partitioned subspecies defined by morphological differences[2]. Analyses of

mitochondrial d-loop variation have suggested that the eastern blue wildebeests are the result of a northward expansion into eastern Africa, following historical confinement to a climate refugium in southern Africa, but did not find genetic structure consistent with the five subspecies[5]. However, the subspecies designation has never been assessed using genomic data, and the evolutionary processes that have led to morphologically distinct blue wildebeest lineages are largely unknown. It is also unknown how the unique ecology of the blue wildebeest, as a migratory and locally superabundant species, has affected the distribution of its genetic variation. Similarly, little is known about the evolutionary processes impacting the lesser-known sister species, the black wildebeest (*C. gnou*), and its evolutionary relation with the blue wildebeest. Although geographically more restricted, the black wildebeest was historically very abundant in southern Africa, with population sizes estimated to be in the hundreds of thousands[6]. But in sharp contrast with the widespread and abundant blue wildebeest, it was reduced to as few as 300 individuals[7] in the early 20th century, due to overhunting and persecution[8]. It subsequently recovered due to a determined conservation effort, breeding programs and reintroductions into southern Africa[8].

The two wildebeest species are able to interbreed[9], although allopatry and different ecological specializations are believed to have prevented widespread natural hybridization[9,10]. However, the lack of complete reproductive isolation has raised concerns about genetic swamping of the black wildebeest by the much more abundant blue wildebeest, threatening the long-term persistence of the black wildebeest[9]. The blue wildebeest has been found to be paraphyletic with respect to the black wildebeest in an mtDNA tree[11], which suggested that introgression may have occurred between the two species. However, the extent, direction and timing of any such introgression have not been estimated, leaving the evolutionary relations between the two species unresolved and limiting a comprehensive conservation management plan for the more vulnerable black wildebeest. In addition, if the two species have admixed, the genomic distribution of ancestry and genetic differentiation would greatly facilitate our understanding of unique adaptations in the two species and whether or not there are parts of the genome that resist introgression despite the presence of gene flow[12].

Finally, because of the superabundance of wildebeest in their optimal habitats and their preference for the same habitats as cattle, sheep and goats, they epitomize the conflicts between humans and wildlife that are occurring in Africa as well as on other continents[13–15]. As a species characterized by seasonal migration in search of pasture and water, the blue wildebeest could be more negatively impacted by an increasingly human-dominated landscape than other species[16]. Notably, this a timely concern since large population declines have occurred in recent decades due to erection of wildlife fences and the increasing density of roads and other human infrastructure that curtail their migration[13,17]. In this respect, blue wildebeest are representative of a more general global trend in which migratory ungulates have been severely affected by human development in the last two centuries[18]. Migration is a key life-history strategy or foraging behavior of many animal species, often enabling them to maintain higher population sizes in spatiotemporally variable environments than they would otherwise[19,20], and enhancing their contributions to ecosystem services[21,22]. Even so, human activities are having an outsized impact on highly migratory species[23,24], including curtailing important migration routes, which can cause population collapse[18,25]. Given its importance, animal migration has been extensively studied from the ecological, behavioral and physiological perspectives[26,27]. However, despite a rich literature on the theoretical role of gene flow in population genetics[28], we know surprisingly little about how different realized migration regimes within a species influence population genetic patterns, and how these patterns may respond to anthropogenically driven migration collapse. Hence, understanding the distribution of genetic variation in a highly abundant and migratory keystone mammal, like the blue wildebeest, and how this is impacted by human-induced habitat fragmentation, is fundamental to the conservation of many large and healthy wildlife populations[18].

In this study, we present population genetic analyses of 121 blue wildebeest and 22 black wildebeest genomes in order to investigate the evolutionary genetics of the wildebeests in light of their unique migratory ecology, and to assess the impact of recent anthropogenic habitat fragmentation on their genetic diversity. We show the genetic structure within the widespread blue wildebeest, and elucidate the historical processes of population divergence, demographic history and gene flow between the two species and between blue wildebeest subspecies. Our results furthermore reveal how genetic variation within wildebeest populations may be shaped by movement behaviors, which have been affected by recent human activities. By comparing our findings to genetic studies of migration in other organisms, we hope to provide new insights into the genomic consequences of disrupting migration routes.

## Results

We generated whole genome resequencing data (average depth: 18.2X, range: 13.1–30.1X, Supplementary Data 1) for 143 wildebeest samples across the two species' ranges, including samples from black wildebeest and all subspecies of the blue wildebeest (Fig. 1A): Brindled (*C. t. taurinus*), Nyassa (*C. t. johnstoni*), Cookson (*C. t. cooksoni*), Eastern (*C. t. albojubatus*) and Western (*C. t. mearnsi*) white-bearded. We mapped and applied rigorous site filtering to chromosome-level genome assemblies of both the blue wildebeest and the domestic goat (*Capra hircus*), resulting in a genomic dataset consisting of 1,540,922,509 and 1,137,510,746 bases, respectively (Supplementary Data 2, for number of SNPs see Supplementary Data 3). We discarded 12 samples due to low DNA quality, first degree relatedness or sample duplication. Downstream analyses were based on genotype calls or genotype likelihoods for the remaining 131 samples according to the requirements of the methods used (see Methods and Supplementary Data 4).

### Genetic structure and phylogeography

We first visualized the population structure using principal component analysis (PCA, Fig. 1B). The first principal component (PC1) distinctly separated the two species, whereas PC2 captured the latitudinal clines within the blue wildebeest. The clustering along PC2 largely reflected the recognized blue wildebeest subspecies, although Cookson and Nyassa clustered very close to each other. Additionally, PC1 and PC2 also indicated some sub-clustering within both the Eastern white-bearded and Brindled subspecies. The population structure revealed by PCA was further supported by a neighbor-joining tree based on identity-by-state, which exhibited a basal split between the black and blue wildebeest (Fig. S1) and partitioned the blue wildebeest individuals according to subspecies. The tree also indicated further clustering of distinct sampling localities within the Eastern white-bearded and Brindled subspecies.

To investigate the population substructure within blue wildebeest further, we inferred ancestry proportions for all the blue wildebeest samples using ADMIXTURE[29]. The highest number of clusters for which ADMIXTURE converged was $K = 12$ (Fig. 1C). When assessing the fit of the ADMIXTURE models using evalAdmix[30] (Figs S2–S3), we found both positive and negative mean correlations of residuals between localites at lower K values, suggesting a poor model fit. At $K = 12$, the Western white-bearded was modeled as one single cluster across four unique localities. In contrast, the Eastern white-bearded and Brindled were subdivided into three and six clusters respectively, consistent with sampling localities. Similar to the long-range panmixia found in Western white-bearded, we identified another long-range panmictic cluster in Brindled, which was labeled as B-Kalahari due to its

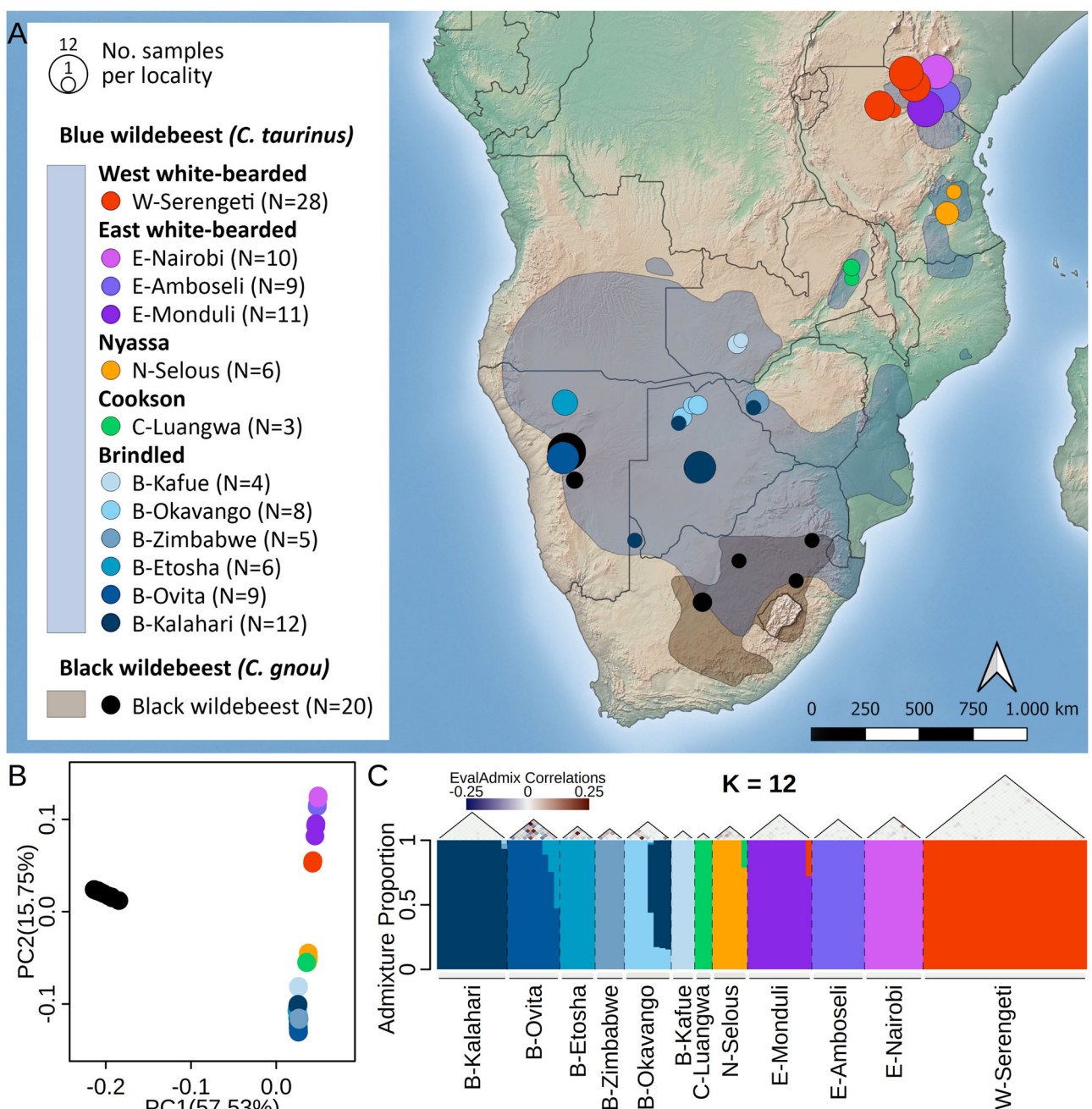

**Fig. 1 | Population structure of wildebeest. A** Origins of samples included in this study. Blue wildebeest samples were grouped into 12 populations inferred by an ADMIXTURE analysis (**C**) and colored accordingly. The sizes of the points that represent the different populations reflect their sample sizes. Populations with samples from the Brindled, Cookson, Nyassa, Eastern and Western white-bearded subspecies are named "B-", "C-", "N-", "E-" and "W-", respectively, followed by the name of their sampling area. The remaining populations are named after the subspecies that they belong to. Shaded species ranges were drawn according to Kingdon[59] and the IUCN Red List. **B** Principal component analysis showing genetic clustering of the 131 wildebeest. Clusters were identified and matched with genetic clusters found in other analyses (**C** and Fig. S1). **C** Barplot illustrating the ancestry proportions inferred for all blue wildebeest samples by ADMIXTURE assuming 12 ancestral populations (*K*). Pairwise correlations of residuals between individuals estimated by evalAdmix are shown above the barplot, while the mean of these correlations of residuals for all pairs within each population is shown below the barplot.

congruence with this geographical feature. Besides samples from central Botswana, this cluster includes one individual from the Okavango area and one from Zimbabwe that were therefore assigned to B-Kalahari (Fig. 1A). The clusters inferred at *K* = 12 were used as population units in all remaining analyses (Fig. 1A). Based on the same analysis, we identified cross-subspecies admixture in one Eastern white-bearded individual from Tanzania (E-Monduli) with some ancestry of the Western white-bearded, and in one Nyassa individual with Cookson ancestry (Fig. 1C). Recent admixture was also inferred in

the Brindled population in northern Botswana (B-Okavango), which showed admixture from B-Kalahari, and in central Namibia (B-Ovita), which showed admixture from B-Etosha. Based on the evaluation of the fit of the ADMIXTURE model (*K* = 12) using evalAdmix and patterns of linkage disequilibrium (LD) decay, we identified nine homogenous populations of blue wildebeest that cover all five subspecies. These populations include B-Etosha, B-Kafue and B-Kalahari from Brindled, C-Luangwa from Cookson, N-Selous from Nyassa, E-Amboseli, E-Monduli and E-Nairobi from Eastern white-bearded, and W-Serengeti

**Table 1 | Hudson's $F_{ST}$ between all wildebeest population pairs**

|             | B-Kafue | B-Kalahari | B-Zimbabwe | B-Okavango | B-Ovita | C-Luangwa | E-Amboseli | E-Monduli | E-Nairobi | N-Selous | W-Serengeti | Black |
|-------------|---------|------------|------------|------------|---------|-----------|------------|-----------|-----------|----------|-------------|-------|
| B-Etosha    | 0.123   | 0.028      | 0.137      | 0.045      | 0.072   | 0.213     | 0.243      | 0.225     | 0.273     | 0.224    | 0.186       | 0.520 |
| B-Kafue     |         | 0.128      | 0.232      | 0.130      | 0.181   | 0.288     | 0.285      | 0.268     | 0.316     | 0.304    | 0.227       | 0.560 |
| B-Kalahari  |         |            | 0.112      | 0.029      | 0.085   | 0.187     | 0.230      | 0.211     | 0.260     | 0.195    | 0.173       | 0.516 |
| B-Zimbabwe  |         |            |            | 0.134      | 0.194   | 0.289     | 0.317      | 0.300     | 0.347     | 0.296    | 0.264       | 0.575 |
| B-Okavango  |         |            |            |            | 0.102   | 0.209     | 0.243      | 0.225     | 0.273     | 0.219    | 0.186       | 0.524 |
| B-Ovita     |         |            |            |            |         | 0.267     | 0.289      | 0.272     | 0.319     | 0.277    | 0.235       | 0.549 |
| C-Luangwa   |         |            |            |            |         |           | 0.312      | 0.293     | 0.345     | 0.209    | 0.260       | 0.615 |
| E-Amboseli  |         |            |            |            |         |           |            | 0.094     | 0.041     | 0.310    | 0.170       | 0.583 |
| E-Monduli   |         |            |            |            |         |           |            |           | 0.135     | 0.288    | 0.150       | 0.575 |
| E-Nairobi   |         |            |            |            |         |           |            |           |           | 0.342    | 0.203       | 0.600 |
| N-Selous    |         |            |            |            |         |           |            |           |           |          | 0.264       | 0.618 |
| W-Serengeti |         |            |            |            |         |           |            |           |           |          |             | 0.554 |

from Western white-bearded (Supplementary Note 1, Figs S2–S6). For the black wildebeest, ADMIXTURE and evalAdmix suggest a diffuse population structure that cannot be explained by discrete clusters (Figs S7–S8), nor by introgression from blue wildebeest, as this was not supported by an ADMIXTURE analysis including both species (Figs S9–S10). However, based on relative genetic homogeneity and geographical coherence, we defined a homogenous black population consisting of a subset of the samples from Namibia (Fig. S11). All subsequent analyses potentially sensitive to substructure were restricted to this subset of black wildebeest samples and the nine homogenous blue wildebeest populations. Although these homogeneous populations have varied sample sizes, the inclusion of populations of small sample sizes should not impact the results since most of the subsequent analyses were either individual based or robust to sample sizes.

To quantify the levels of genetic differentiation between the inferred populations, we used Hudson's $F_{ST}$ (Table 1). $F_{ST}$ between the blue and black wildebeest populations ranged from 0.52 to 0.62. $F_{ST}$ values between populations from the different subspecies of blue wildebeest were moderate (0.15–0.35) compared to the $F_{ST}$ between populations within subspecies (0.03-0.23). Notably, the Western white-bearded population, W-Serengeti, and the Brindled population in Kalahari, B-Kalahari, exhibited one of the lowest between-subspecies differentiation (0.17), despite their large geographic distance of separation.

## Population histories and genetic diversity

To assess changes in historical effective population sizes we conducted PSMC analysis[31], which indicated that the two species diverged no later than 500 kya (Fig. 2A). Following the species split, all blue wildebeest shared a very similar trend of population size until 150 kya. After that, the Eastern and Western white-bearded wildebeest experienced moderate population expansion followed by a more recent reduction (between 10 and 50 kya) to lower modern population sizes. Notably, all the Brindled populations showed signs of continuously increasing effective population sizes starting from 150 kya until 20 kya, which could, however, be caused by introgression (see below). Since PSMC has limited power in inferring recent demographic changes (<≈20 kya), we investigated the recent population history of wildebeest by applying PopSizeABC[32], which incorporates the SFS and patterns of LD, to the populations with sufficient sample size (Fig. 2B). The black wildebeest showed a prolonged drastic population contraction in the past 1000 years with the effective population size decreasing from around 10,000 to only 100 individuals, in line with the historical record of near extinction in recent centuries. Such a signal was not observed in any of the blue wildebeest populations.

To further characterize recent demographic history, we inferred runs of homozygosity (ROH) and genome-wide heterozygosity for each wildebeest sample. In general, ROH and heterozygosity show a very strong correlation within each population (Fig. S12). Nearly all the black wildebeest samples had large tracts of ROH (>10 Mb, Fig. 2C), suggesting high levels of inbreeding within the last ≈25 generations. This recent inbreeding in black wildebeest is likely caused by the severe 19th-20th century bottleneck followed by persistence in artificially small, managed populations. In contrast, the blue wildebeest populations showed a range of different ROH patterns. The two Brindled populations, B-Kalahari and B-Etosha, and the Western white-bearded population, W-Serengeti, had almost no ROH, whereas most other blue wildebeest populations exhibited substantial ROH proportions, though lower than those observed in the black wildebeest. The genome-wide heterozygosity of blue wildebeest ranged between 0.0017–0.0027 (Fig. 2D), which is approximately half that of e.g., Cape buffalo[33] and waterbuck[34]. Interestingly, the black wildebeest did not show significantly reduced genetic diversity relative to its abundant sister species, instead, the lowest heterozygosities were observed in the Cookson and Nyassa subspecies. The Brindled wildebeest populations had very variable heterozygosities, with B-Kalahari and B-Etosha presenting the highest values. To investigate how heterozygosity was impacted by recent inbreeding, we re-estimated heterozygosity based on the genomic regions outside the ROH (Fig. 2E). After removing these regions, populations within subspecies have a very similar baseline heterozygosity (as expected in a panmictic population), indicating recent inbreeding as the main driver of the observed within subspecies variability in genome-wide heterozygosity (Fig. 2D). Interestingly, re-estimated heterozygosity of the black wildebeest is comparable to some of the highest values in the blue wildebeest, suggesting that recent inbreeding has markedly reduced genetic diversity of the former.

## Ancient hybridization in southern Africa

To test for past gene flow among the wildebeest populations, we calculated the $F$-branch statistic using Dsuite[35] (Fig. 3A). We found pervasive signals of interspecies admixture between all Brindled populations and the black wildebeest. In addition, we observed high $F$-branch values between different subspecies of the blue wildebeest, suggesting gene flow on the intraspecific level. Surprisingly, these signals were strongest between the Nyassa population, N-Selous, and the Cookson population, C-Luangwa, and between the two most geographically distant Brindled populations, B-Kalahari and B-Etosha.

To corroborate and summarize these gene flow events, we explored admixture graphs using qpGraph[36] with a subset of the populations, allowing for up to 5 migration events. The best fitting admixture graph (Fig. 3B) had 4 migrations and was significantly

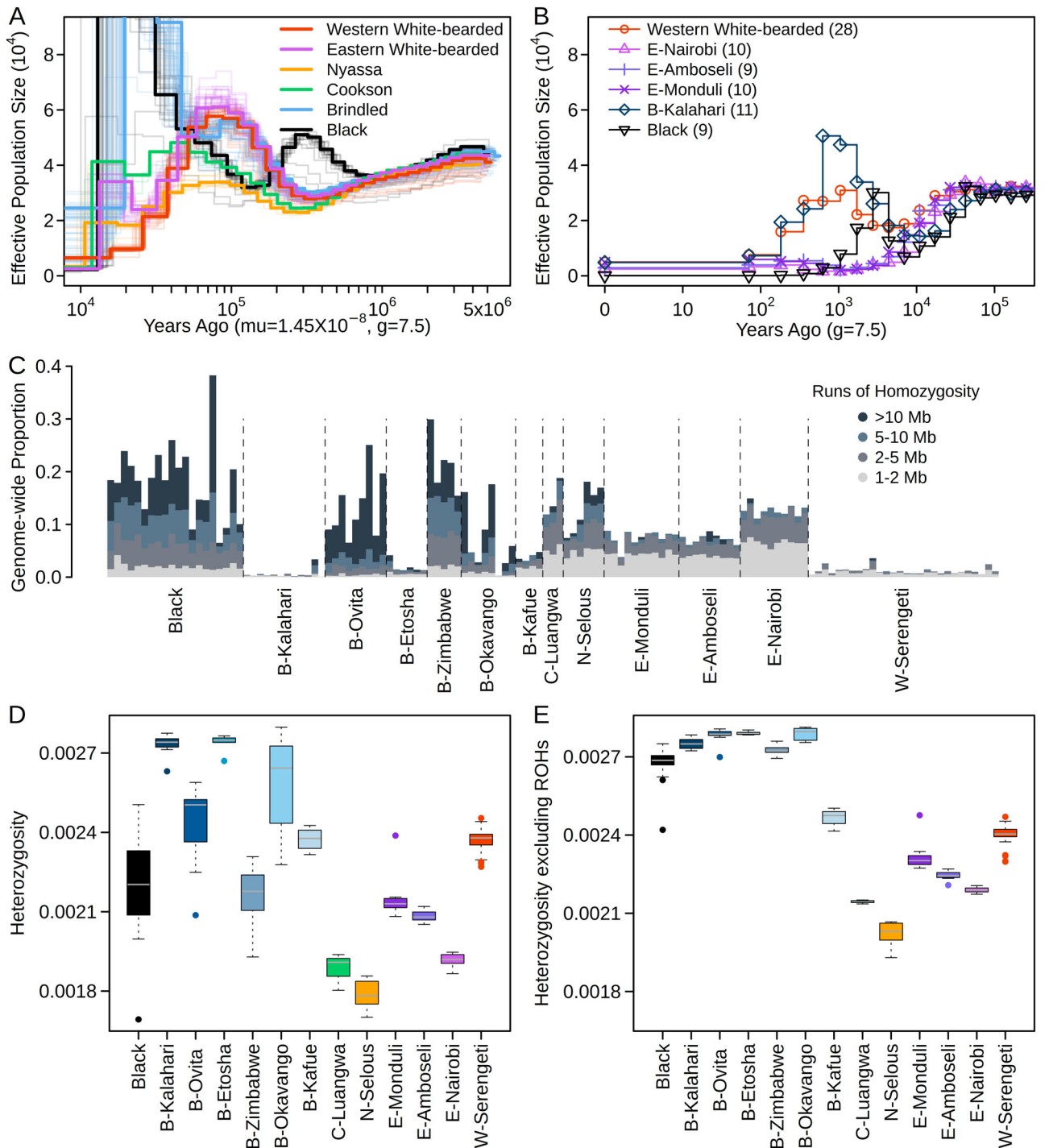

**Fig. 2 | Demographic histories, inbreeding and heterozygosity. A** Effective population sizes over time of all wildebeest samples estimated using PSMC. **B** Recent effective population sizes of the populations with sufficient sample size ($n \geq 9$) inferred using PopsizeABC. **C** Estimated proportion of runs of homozygosity (ROH) per individual genome. **D** Genome-wide heterozygosity measured as proportion of heterozygous sites in the genome. **E** Measures of genome-wide heterozygosity excluding ROHs. Sample sizes of the populations included in (**D**, **E**) are as follows: Black ($n = 20$), B-Kalahari ($n = 12$), B-Ovita ($n = 9$), B-Etosha ($n = 6$), B-Zimbabwe ($n = 5$), B-Okavango ($n = 8$), B-Kafue ($n = 4$), C-Luangwa ($n = 3$), N-Selous ($n = 6$), E-Monduli ($n = 11$), E-Amboseli ($n = 9$), E-Nairobi ($n = 10$), W-Serengeti ($n = 28$). Boxplots in (**D**, **E**) indicate median (center line), the 25th and 75th percentiles (box), and the highest and lowest values within the upper and lower quartiles ±1.5* interquartile range (IQR), respectively (whiskers).

($p < 0.05$) better than all graphs with 0 to 3 migrations, while not significantly worse than any graph with 5 migrations. However, 20 other graphs with four migrations were not significantly worse (Fig. S13). Importantly, all the 21 non-rejected graphs with four migrations incorporated the same migration event from black wildebeest into the southern population of blue wildebeest (B-Etosha) with the admixture proportion varying from 6% to 14%, with 12% in the best scoring graph (Supplementary Data 5). The best scoring graph furthermore identified introgression between blue wildebeest subspecies, including 8% admixture from Brindled into Cookson, and a similar amount of admixture from the sister branch of Nyassa into Eastern white-bearded (9% admixture into E-Monduli, and 7% admixture into the ancestor of

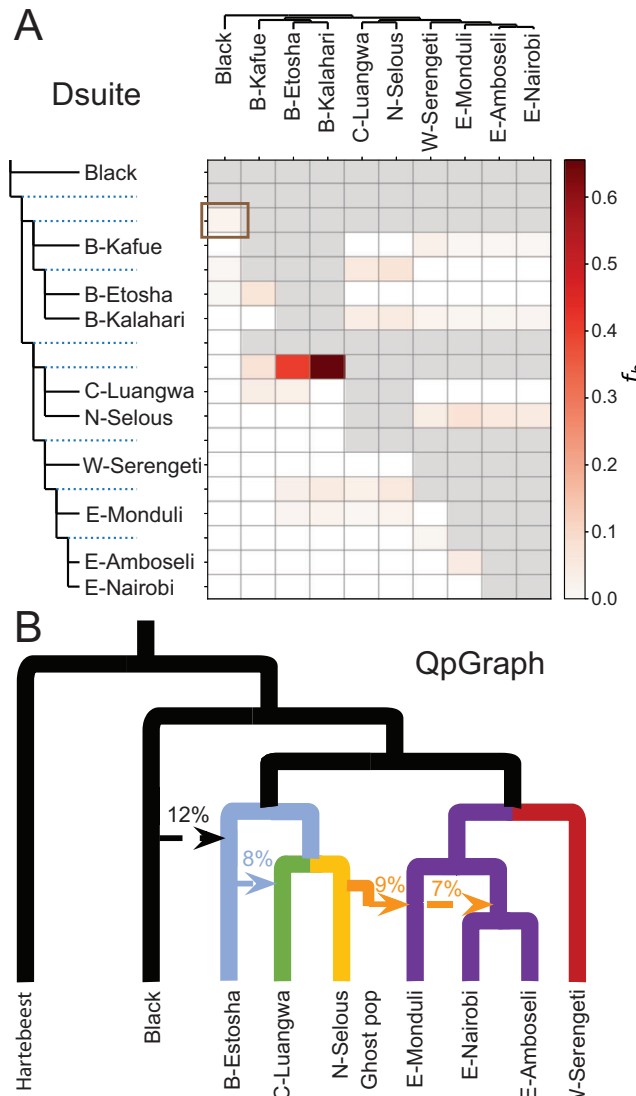

**Fig. 3 | Main gene flow events in wildebeest. A** Heatmap of *F*-branch ($f_b$) statistics for non-admixed wildebeest populations using Dsuite. Calculation of $f_b$ statistics was constrained to the groupings of populations that fit with the supplied population tree (see Fig S1), which is shown along the y axis. Each branch of the tree, including the internal branch (indicated by blue dashed line) that represents the ancestor population of branches below, points to a corresponding row in the matrix with inferred $f_b$ statistics. The value in the matrix measures the extent of allele sharing between the corresponding branch of the population tree on the y axis (relative to its sister branch) and the population on the x axis (known as P3 in standard *D*-statistics). Gray color in the matrix means that calculation of $f_b$ statistics is not applicable given the topology of the population tree. To illustrate, the highlighted cell (in dark brown rectangle) indicates significantly stronger allele sharing between the black wildebeest (x axis) and the ancestor of all included Brindled populations, represented by the internal branch above populations B-Kafue, B-Etosha and B-Kalahari, relative to its sister, the ancestor of all the other blue wildebeest populations, represented by the internal branch above populations Cookson, Nyassa, W-Serengeti, E-Moduli, E-Amboseli and E-Nairobi. **B** The highest-scoring admixture graph with four admixture events inferred by qpGraph. Percentages above the arrows (admixture edges) show the admixture proportions. The four ancient gene flow events include: (1) introgression from black wildebeest into B-Etosha, (2) introgression from B-Etosha into Cookson, (3) introgression from a ghost population as a sister branch of Nyassa into E-Moduli, and (4) introgression from the ghost population into the ancestor of E-Nairobi and E-Amboseli.

E-Nairobi and E-Amboseli). However, these intraspecific events were not uniformly supported by all the 21 non-rejected graphs (Supplementary Data 5) and are therefore less certain.

We further investigated the interspecific gene flow using outgroup f3 statistics (Fig. 4A) and observed a slight variation across the range, with the Brindled population in northern Namibia (B-Etosha) exhibiting the highest black wildebeest affinity and the Brindled population in Zambia (B-Kafue) the lowest. Because the inferred direction of gene flow from black to the Brindled populations rather than the opposite direction was unexpected, we tried to validate it using different approaches. First, we estimated the polarized frequency spectrum of *D*-statistics ($D_{FS}$[37]). The peak among low-frequency bins in $D_{FS}$ supported the notion that introgression predominantly occurred from the black wildebeest into the Brindled wildebeest (Fig. S14). Gene flow between black and Brindled wildebeest was furthermore supported by the mitochondrial DNA phylogeny, which places all Brindled wildebeests on a sister branch to all black wildebeests, to the exclusion of all the other blue wildebeests (Fig. S15).

To corroborate the interspecific introgression and confirm that it did not occur recently, we estimated the distribution of lengths of introgression tracts by inferring local ancestry in the Brindled wildebeest population (B-Etosha), relative to N-Selous, a population that did not show signs of interspecific introgression in the above-mentioned analyses. We used the black wildebeest and the Western white-bearded wildebeest as reference populations for pure black and blue wildebeest ancestry, respectively. We detected around 5% black wildebeest ancestry in the B-Etosha individuals and a negligible amount in the N-Selous individuals (Fig. 4B, S16–17), confirming the relative amounts of introgression inferred above. An absence of large tracts of black wildebeest ancestry (only one fragment >1 Mb) in any B-Etosha individual indicates that the introgression from the black wildebeest is highly unlikely to have occurred within the past 100 generations, as we would expect a fairly large proportion of admixture tracts spanning >1 cM assuming a pulse admixture event of 0.1 occurring 100 generations ago according to simulations by Liang and Nielsen (2014)[38]. We next used fastSimcoal2[39] to estimate time of the interspecific admixture using a demographic model with a fixed single admixture pulse of 12% from black wildebeest to B-Etosha in accordance with the highest scoring qpGraph. This led to an estimate of around 4210 generations or ≈32 kya (95% confidence interval: 23–40 kya, Fig. 4C, Supplementary Data 6). We also used fastSimcoal2 to investigate whether the gene flow can be better modeled as continuous introgression as opposed to a single pulse of introgression. The likelihoods for the different models were very similar (Fig. S18), suggesting that we cannot reliably distinguish between the two types of introgression.

Given the substantial ancient gene flow between species, we were interested in identifying regions of the genome that might resist introgression between the two wildebeest species. For this purpose, we investigated genetic landscapes of differentiation between the black wildebeest and Brindled wildebeest (B-Etosha). A genome scan identified a large block of elevated $F_{ST}$ on chromosome 1, spanning ≈7 Mb (Figs S19-20). The $F_{ST}$ peak also exhibited local reduction in nucleotide diversity and significantly stronger LD compared with the remaining regions on chromosome 1 (Fig. S20) in both species. Pairwise alignment between the blue wildebeest genome and domestic goat genome indicated that chromosome 1 of wildebeest was formed by the fusion of two ancestral chromosomes, which was also reported in Vozdova et al.[40] Intriguingly, the region of $F_{ST}$ peak spans the fusion site (Fig. S21) and harbors genes related to coat color or skin pigmentation such as *OCA2* and *HERC2*[41], as well as oxygen metabolism – hemoglobin gene clusters including *HBQ*, *HBM1*, and *HBZ1*[42].

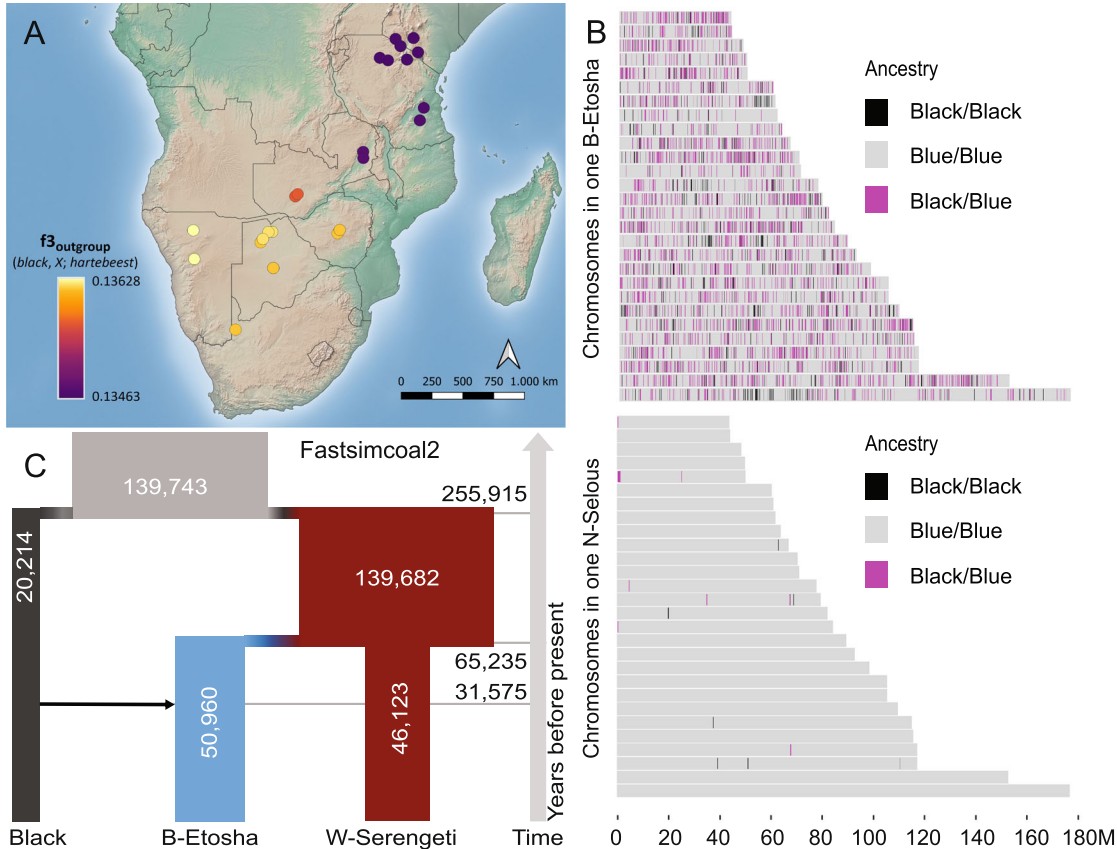

**Fig. 4 | Evolutionary relation between the black and the blue wildebeest.**
**A** Shared evolutionary history between the black wildebeest and different blue wildebeest populations estimated using outgroup f3 statistics in the form of f3 (black, $X$; hartebeest), where $X$ denotes different populations of blue wildebeest shown on the map. **B** Inferred local ancestries in samples of the Brindled wildebeest from Etosha (upper) and Nyassa wildebeest (lower) using the black wildebeest and the

Western white-bearded wildebeest as the ancestor populations. **C** The demographic model and point estimates of demographic parameters in fastsimcoal2, where a one-pulse admixture of 12% from the black wildebeest to the Brindled wildebeest was fixed. Numbers within the bars are estimated effective population sizes. Numbers at the horizontal gray lines are estimated time (in years) for the corresponding demographic event.

## Contrasting genomic signatures between migratory and non-migratory populations

To investigate how migration impacts the genetic variation in blue wildebeest, we compared spatial patterns of genetic structure, demographic history, levels of genetic diversity and inbreeding between populations with long-distance migrations (migratory) and populations with sedentary or vagrant behaviors (non-migratory) in eastern (Eastern and Western white-bearded; Fig. 5A) and southern Africa (Brindled; Fig. 5B, Supplementary Note 2). These analyses point to striking and consistent differences between migratory and non-migratory populations. EEMS analyses[43] revealed that in both regions, migratory populations had fewer barriers to gene flow across large geographical distances (Fig. 5C, D). For example, no gene flow barriers were detected in the entire Serengeti-Mara region where the Western white-bearded were sampled, while the three non-migratory populations of the Eastern white-bearded were clearly genetically disconnected (Fig. 5C). Additionally, the migratory populations in both eastern and southern Africa were also characterized by larger effective population sizes reflected in faster decay of LD (Fig. 5E, F), which is consistent with the popsizeABC results (Fig. 2B). Finally, the migratory populations generally had higher heterozygosity levels and smaller ROHs proportions (Fig. 5G-H). The close correlations between heterozygosity levels and ROH (Fig. S12) suggest that recent inbreeding—possibly related to differences in migratory behavior, may have been instrumental in shaping the current genetic diversity. Of note, we found an unexpected pattern of heterozygosity and ROH in B-Kafue,

which deviated from the linear relationship observed in Brindled wildebeest. B-Kafue also received less introgression from the black wildebeest compared to the other Brindled populations (Fig. 3A) and was located in the region surrounded by strong barriers to gene flow as shown in EEMS (Fig. 5B). This indicates a distinct evolutionary history of B-Kafue relative to the other Brindled populations.

## Discussion

Our study represents a comprehensive population genomic analysis of wildebeests, offering new insights into their evolutionary history and on population genetics in a highly migratory species. First, in contrast to previous studies using mtDNA, our whole-genome data clearly support the previously defined wildebeest subspecies[44] as meaningful evolutionary units. Hence, the current blue wildebeest subspecies classification reflects the evolutionary history of the species, unlike for some other African mammals for which genetic analyses have been in partial conflict with subspecies classification schemes[45–49]. However, with moderate $F_{ST}$ values (up to 0.35) and relatively recent divergence times (65 kya), blue wildebeest subspecies are not highly differentiated or deeply divergent lineages.

Second, the black and blue wildebeest are known to hybridize and produce fertile hybrid offspring when they come into contact throughout South Africa[50]. As a rare and geographically confined species, the black wildebeest is more vulnerable to depletion of its unique genetic variation or even to extinction due to hybridization[51,52]. Previous studies based on microsatellites reported evidence of

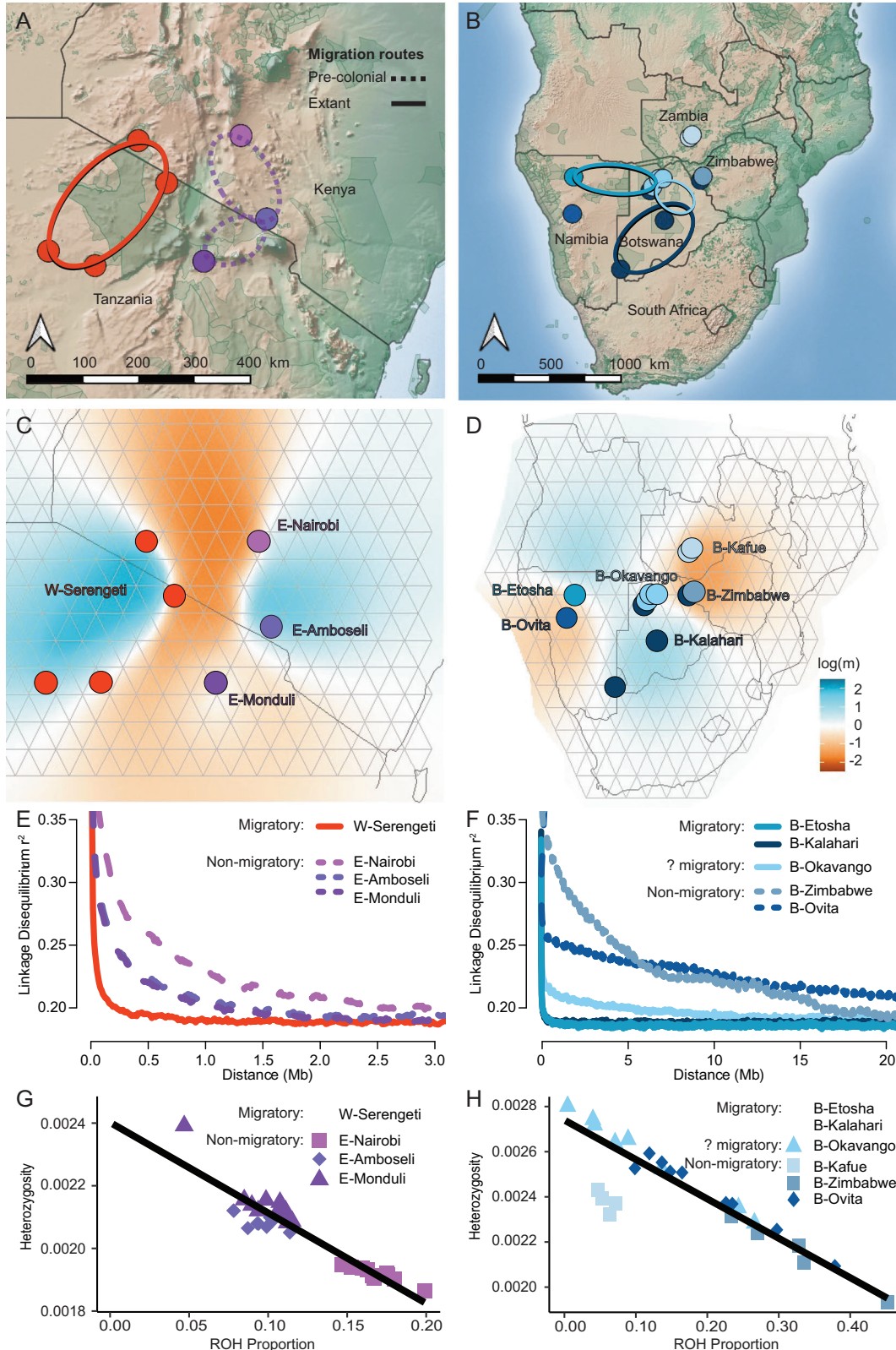

**Fig. 5 | Comparison between adjacent migratory and non-migratory populations in eastern (left) and southern Africa (right).** Sampling locations with recorded historical (dotted line) and extant (solid line) migration paths for the white-bearded wildebeest (**A**) and Brindled wildebeest (**B**) based on Msoffe et al.[105] **C**, **D** Estimated effective migration surfaces for the migratory Western white-bearded population and non-migratory Eastern white-bearded populations using EEMS. Note that localities of populations are anchored to the grid used for EEMS; true GPS localities are depicted in (**A**, **B**). **E**, **F** Decay of linkage disequilibrium measured as mean r² for SNP pairs stratified by the genomic distances for populations (n > 5). Linear relationships between the proportions of runs of homozygosity (ROH) and levels of genome-wide heterozygosity in the white-bearded wildebeest (**G**) and Brindled wildebeest (**H**).

introgressed alleles from the blue wildebeest in the black wildebeest in South Africa[53]. Our results, however, suggest a more complicated pattern of interspecies admixture in which the most important feature is an ancient (≈32 kya) introgression of black wildebeest ancestry into the southernmost populations of blue wildebeest. Gene flow between black and Brindled wildebeest can explain the paraphyly of the blue and black wildebeest, which was previously observed in mtDNA phylogenies[11] and further corroborated here. In contrast, we conclude that recent admixture between the two species, though previously documented, cannot be widespread, as we found no indications of it in our data set including black wildebeest from across its natural and introduced ranges. The ancient gene flow between the two wildebeest species inferred here is consistent with a Pleistocene range expansion of the widespread blue wildebeest, which may have brought the two previously allopatric wildebeest species into contact in southern Africa. Hybridization resulting from range expansion has been shown to lead to considerable amounts of introgression in the expanding species, but very little signal of introgression into the stationary species[54,55]. Hence, we show that rather than the hypothesized pattern of recent introgression from blue into black wildebeest, the Brindled wildebeest carries the legacy of historical, possibly late Pleistocene, introgression from black wildebeest, best explained by a spatial expansion of the Brindled wildebeest. The ancient introgression date and apparent lack of recent gene flow highlights the dynamic nature of wildebeest phylogeography, consistent with climate change playing an important role in shaping both distribution and gene flow patterns in wildebeest, as in many other African ungulates[56,57]. It is not possible to fully exclude a limited amount of introgression in the opposite direction, but our results clearly suggest that at the very least gene flow was strongly asymmetric. Lastly, the complete lineage sorting of all black wildebeest mtDNA haplotypes as sister to all Brindled mtDNA haplotypes, although consistent with the above introgression scenario, is puzzling and could suggest either introgression in the other direction, repeated pulses of introgression over longer time periods, extreme genetic drift or even selection. Despite considerable historical gene flow between the black and Brindled wildebeest, we identified a single region on wildebeest chromosome 1 with highly elevated genetic differentiation, low genetic diversity and increased LD in each population. Such a region could potentially harbor loci underlying differential adaptations or loci involved in reproductive isolation between the two species. Indeed, we found several genes with interesting phenotypic effects, such as color coding genes (OCA2, HERC2) and a hemoglobin gene cluster in this region. However, it also spans a chromosome fusion site in the wildebeest genome, and reduced recombination could therefore be related to the fusion event, facilitating more efficient background selection[58]. Consequently, the altered recombination landscape caused by the ancestral chromosome fusion could have contributed to this region of elevated between species differentiation.

Third, we found very marked and consistent genetic patterns when contrasting migratory with non-migratory populations in either extreme of the blue wildebeest range. Migratory populations exhibit shared genetic characteristics, including 1) increased genetic connectivity over extended geographical areas, 2) higher genetic diversity, 3) lower level of inbreeding, and 4) faster LD decay suggesting higher recent population sizes relative to neighboring non-migratory populations. All of these suggest that the migratory populations have maintained larger effective population sizes and genetic homogeneity over larger geographical areas than their neighboring non-migratory populations, leading to a tangible difference in genetic variation. This provides the first clear evidence that non-migratory wildebeest populations display genetic features that are usually considered negative from a conservation genetic perspective, and that these are absent in migratory wildebeest populations. It suggests that long-range migration is a natural condition for blue wildebeest populations throughout the species' range, and that disruption of such migration

would almost certainly have negative genetic consequences. Concordantly, many of the wildebeests' unique adaptations are related to the ability to stay on the move[59]. For example, wildebeest calves can walk and run unassisted within minutes of birth, which is rare among ungulates and allows wildebeests to calve shortly before undertaking their characteristic long-range annual migrations[60]. However, many wildebeest migrations have been severely disrupted by human disturbances[18,61], and the curtailing of established migration routes is known to have caused severe declines or extirpation of some populations[23,44]. As a case in point, the Western and Eastern white-bearded wildebeest populations had historically similar large-scale annual migrations, however, these have been increasingly disrupted by human activities, especially for the Eastern white-bearded wildebeest, since the beginning of European colonialism in the late 1800s[62,63] (Supplementary Note 2). Consequently, these two formerly comparable subspecies now show very distinct genetic signatures (Fig. 5A, E, G).

The ubiquity of these genetic features in geographically separated migratory wildebeest populations is consistent with observations across a spectrum of migratory species including mammals, birds and insects. For example, the Barren-ground ecotype of reindeer (Rangifer tarandus groenlandicus), engaging in large-scale and long-distance migrations, clearly forms a more uniform genetic cluster compared to other sedentary or partially migratory ecotypes in North America[64]. Similarly, in European blackcaps (Sylvia atricapilla), a widely distributed songbird, populations that migrate over medium to long distances exhibit increased genetic homogeneity and reduced genetic differentiation[65]. Moreover, among 97 Lepidoptera species, the migratory species show greater genome-wide heterozygosity than the non-migratory species[66]. Collectively, these findings suggest the presence of a "population genetic migratory syndrome", conceptually similar to the broader "migratory syndrome" initially proposed by Dingle[67] to describe convergent evolution of phenotypic traits that differentiate non-migratory from migratory forms. This finding raises concerns that the disruption of migration routes in naturally nomadic species leads to genetic erosion and long-term loss of evolutionary potential, in addition to the well-known short-term ecological impact[23,24].

Fourth, given the high density of blue wildebeest across its distributional range, one unanticipated finding was its modest genetic diversity and small effective population size. Heterozygosity in the blue wildebeest ranged between 0.0017–0.0027. For comparison, the median heterozygosity for 35 species of Bovidae, most of which have lower or much lower population sizes than wildebeest, was 0.0021[68]. Although the 19th century Rinderpest pandemic caused high mortality of up to 95% in wildebeest[69], this is unlikely to explain the relatively low genetic diversity, as we did not find evidence for a sudden and recent decrease in effective population size across the range (Fig. 2B). Furthermore, Rinderpest had a similar mortality rate in Cape buffalo (Syncerus caffer caffer) without any apparent reduction in its genetic diversity[33]. Instead, low genetic diversity in wildebeests could be related to skewed reproductive success in wildebeest males[70] or other life history traits. Additionally, contrary to expectations for a population that recently survived a bottleneck, the black wildebeest exhibits a similar level of genetic diversity compared to many blue wildebeest populations (Fig. 2D, E). This finding might be attributed to the historically high local abundances of black wildebeest[71].

Altogether, this work sheds light on how two major evolutionary forces—hybridization and migration—drive the distribution of genetic variation in wildebeest. Surprisingly, we found that introgression between the two wildebeests is predominantly in the direction of the blue wildebeest, and likely occurred naturally as a result of a late Pleistocene spatial expansion in the more widespread blue wildebeest. This result contrasts with the feared genetic swamping of black wildebeest[52]. We also found that curtailed migration routes are very

likely to lead to negative impacts on wildebeest genetic variation through disruption of metapopulation networks and the resulting decrease in effective population size. It is well known that a reduction in genetic diversity and increase in inbreeding is harmful for natural populations[72]. We therefore emphasize the importance of maintaining intact migration routes for highly migratory ungulates such as wildebeests. This should be taken into account when designing conservation areas and management strategies for populations of wildebeest and other migratory species, including in evaluating the ecosystem costs of building new infrastructure in key wildebeest migration areas such as the Serengeti-Mara, as was recently proposed[73,74]. It remains to be seen whether the genetic migratory syndrome can be fully restored by reinstating migration routes in areas where they have been disrupted. More broadly, our findings are potentially relevant for other migratory ungulates, which may face similar risks of genetic deterioration as habitats become degraded, fragmented or lost and large gene flow networks spanning hundreds of kilometers collapse.

## Methods

### Sampling, DNA extraction, and sequencing

The research presented in this study complies with all relevant ethical regulations and was conducted in accordance with the Code of Conduct for Responsible Research of the University of Copenhagen. Initially, 143 wildebeest tissue samples including 22 black wildebeests and two hartebeest (*Alcelaphus buselaphus*) tissue samples were selected from existing collections at the University of Copenhagen and at the United States Department of Agriculture (USDA). Collection of tissue samples at the University of Copenhagen was carried out in the 1980s and 1990s. The criteria for sample selection included reliability of information on sampling localities and dates, and coverage of species distribution.

The University of Copenhagen tissue samples of wildebeest and hartebeest were stored at −80 °C since the time of collection. DNA of these samples was extracted using the QIAGEN DNeasy Blood and Tissue Kit (QIAGEN, Valencia, CA, USA), following the manufacturer's protocol. RNase was added to all DNA extractions to remove RNA. All extractions were verified for the presence of a high-molecular-weight band using gel electrophoresis and concentrations were measured using a Qubit 2.0 Fluorometer and Nanodrop. Library preparation and DNA sequencing was conducted at GrandOmics, China. Genomic DNA for each sample was randomly fragmented, and then size selected to around 330–530 bp insert size. The fragments were subjected to end-repair and then 3′ adenylated. Adapters were ligated to the ends of these 3′ adenylated fragments, and PCR was performed to amplify the ligated fragments. Next, single-stranded PCR products were produced via denaturation, and circularized. Only single-stranded circular products were retained, while non-circular (linear) DNA molecules were digested. Prior to pooling libraries for sequencing, DNA concentration and library fragment lengths were checked. Processed DNA was sequenced with 150 bp paired-end reads using MGISEQ-T7 technology.

The USDA hide samples of wildebeest were salted, acidified and dried after collection in the field and imported to the USA with a private taxidermy permit. Samples were purchased by the USDA from willing sellers and stored at −20 °C until DNA extraction by standard phenol/chloroform procedures. DNA was dissolved in a solution of 10 mM TrisCl, 1 mM EDTA (TE, pH 8.0) and stored at 4 °C. Sample quality and concentrations were measured by ultraviolet spectrophotometry and double-stranded DNA fluorometry (DeNovix Inc., Wilmington, DE USA; QuantiFluoONE, Promega, Madison, WI, USA). For whole genome sequencing, 2 μg of genomic DNA was fragmented by focused-ultrasonication and used to make indexed, 500 bp, paired-end libraries according to the manufacturer's instructions (TruSeq DNA PCR-Free LT Library Preparation Kits A and B, Illumina, Inc., San Diego, California USA). Pooled, indexed libraries were sequenced with massively parallel sequencing machines (either NextSeq500 or NextSeq2000, Illumina Inc.) and the appropriate kits producing 2 × 150 bp paired-end reads. Samples were repeatedly sequenced to a threshold of 40 gigabases surpassing Q20 quality. This approach produced at least 12-fold mapped read coverage (15-fold average) and provided genotype scoring rates and accuracies that exceed 99%[75].

### Mapping

Sequencing data for all samples were mapped to the reference genomes of both the blue wildebeest (*C. taurinus*, Genbank ID: GCA_006408615.1) and domestic goat (*Capra hircus*, Genbank ID: GCF_001704415.2) with a customized version of PALEOMIX pipeline[76]. As a first step, raw sequencing reads were trimmed for universal Illumina adapter sequences and adapter sequences recommended for the MGI platform using AdapterRemoval v2[77]. Read pairs with overlapping sequences of at least 11 bps were merged in order to increase the fidelity of base calls and specificity in alignment. Contradictory overlapping bases with equal quality scores in the overlapping region were masked as 'N' via the --collapse-conservatively option of AdapterRemoval. Empty reads resulting from primer-dimers were discarded. All the processed reads were mapped using BWA "mem" v0.7.17-r1188[78]. Alignments were sorted and MD tags updated using Samtools v1.11[79] commands "sort" and "calmd", respectively. Duplicates were marked by Samtools "markdup" for unmerged reads and PALEOMIX "rmdup_collpased" for merged reads.

After mapping, we filtered the resulting BAM files using standard BAM flags to exclude unmapped reads, reads with unmapped mates, duplicate reads, QC failed reads and secondary or supplementary alignments. We further discarded alignments with an estimated insert size less than 50 bp or larger than 1000 bp, and alignments where fewer than 50 bp or less than 50% bases in either read were mapped. Additionally, we discarded paired reads mapped to different scaffolds or mapped in improper orientations (e.g., both on the positive strand).

### Data quality filtering

**Genome site filtering.** We applied multiple filters to exclude loci that are prone to mapping or genotyping errors in the reference genomes. First, we identified low complexity and repetitive sequences using Repeatmasker v4.1.1[80] with RMBlast as the search engine and the following settings: -frag 50000 -species Mammalia. Second, we calculated the global depth across all the blue wildebeest samples for each site using Samtools[81]. Based on the resulting distribution of global depths, we removed regions with both extremely high depths (above 1.5 times the median of global depths) and low depths (below half the median of global depths). Third, we filtered out potential paralogs based on excessive heterozygosity, which likely arises due to reads from paralogous loci being mis-mapped to a single location. To do so, we calculated per-site inbreeding coefficients ($F$) using only the blue wildebeest samples based on preliminary biallelic genotype calls with PCAngsd[82,83]. Windows of 10 kb surrounding the sites exhibiting excessive heterozygosity ($F < -0.90$) and significant deviations from Hardy-Weinberg equilibrium ($p < 10^{-6}$) were excluded. Finally, we excluded the X chromosome identified using SATC[84] and short scaffolds (<0.25 Mb) that fail to be anchored to any chromosome. Downstream population genetic analyses (excluding sample quality filtering below) were restricted to the genomic regions that passed all the above-mentioned filters unless stated otherwise.

**Sample quality filtering.** We identified duplicated or closely related samples based on KING-robust kinship[85], which can be used to detect close familial relatives without estimating population allele frequencies. To calculate KING-robust statistics, we estimated two-dimensional site frequency spectra (2d-SFS) for individual pairs with genotype likelihoods using the GATK model (-GL 2) in ANGSD[86]. Based on a KING-robust threshold of 0.375, we identified six pairs of duplicates, and based on a threshold of 0.2, we identified four pairs of first

degree relatives. For each of these pairs, we excluded the sample with lower sequencing depth for downstream analyses.

We next excluded samples with extraordinarily high levels of heterozygosity as these samples very likely suffer from DNA contamination or considerable sequencing errors. To calculate individual heterozygosity, we estimated 1d-SFS with genotype likelihoods using the GATK model in ANGSD. The analysis revealed two individuals with excessively high heterozygosity (≥0.004), which were excluded in downstream analyses. Taken together, 131 wildebeest samples were retained after sample quality filtering.

## Genotype calling and imputation

To fulfill the specific requirements of different analyses, we prepared four different genotype datasets for the downstream analyses (for an overview see Supplementary Data 3). In short, dataset1 is the basic dataset used for analyses of population structure and consists of imputed common variants mapped to the blue wildebeest genome. Dataset2 is intended for PSMC inference and individual heterozygosity estimation. For this dataset, we include genotypes at both variable and non-variable sites mapped to blue wildebeest genome, where we masked genotypes covered by less than eight reads or by individual depth being more than two times its mean depth, as well as heterozygous genotypes supported by less than two reads for any of two alleles. Dataset3 is used for ROH estimation, where we want to minimize the miscalling of heterozygous genotypes, and therefore it contains more strictly filtered common variants mapped to blue wildebeest genome. Specifically, we retained genotype calls with at least 10 reads and heterozygous genotypes supported by at least three reads for each allele and filtered out sites with MAF < 0.05. Dataset4 is used for introgression related analyses where reference bias could be a concern, and thus consists of SNPs mapped to the goat genome, where we only retained genotype calls with at least 10 reads and heterozygous genotypes that were supported by at least two reads for each allele.

For three of the datasets (dataset1, dataset3 and dataset4), we used bcftools v.1.13[79] to jointly call genotypes based on the BAM files of the wildebeest samples. We used the "--per-sample-mF" flag in bcftools mpileup, and the "--multiallelic-caller" flag in bcftools call. All runs of genotype calling were limited to the genomic regions reserved after the site filtering and relied on reads with mapping quality scores of at least 30 and bases with quality scores of 25. Multiallelic sites and indels were excluded in genotype calling. We also performed joint genotype calling for the two outgroup samples of hartebeest with the same settings and merged the resulting genotype files with those of wildebeest samples mapped to the same reference. We only retained sites that are polymorphic in the wildebeest samples in the merged file.

For dataset2, we performed genotype calling individually for each BAM file of the wildebeest samples using bcftools v.1.13. Like the joint genotype calling, the analysis was restricted to the genomic regions retained after the site filtering and based on reads with mapping quality scores of at least 30 and bases with quality scores of 25. We excluded indels in genotype calling.

The imputation for dataset1 was based on the genotype likelihoods at bi-allelic sites of the blue wildebeest genome, which were estimated in bcftools. Imputation was performed using BEAGLE v3.3.2[87] separately for each chromosome. Imputed SNPs with allelic $r^2 < 0.99$ and minor allele frequency (MAF) < 0.05 were filtered out for downstream analyses.

## Population structure

We investigated the population structure of wildebeest using multiple methods based on the imputed common SNPs (dataset1). We first performed principal component analysis (PCA) for all the 131 wildebeest samples using the full SVD from PLINK v1.9[88]. We also estimated admixture proportions for all samples using ADMIXTURE

with a set of randomly selected one million SNPs after pruning based on linkage disequilibrium (LD, $r^2 > 0.6$). Additionally, we constructed a neighbor-joining (NJ) tree based on the covariance matrix of identity-by-state (IBS) distances, which were calculated with the command "--distance" in PLINK. To better understand the population structure within each species, we also estimated admixture proportions for the blue wildebeest and black wildebeest separately using ADMIXTURE with the assumed number of ancestral populations (K) ranging from 2 to 14 and from 2 to 4, respectively. To improve computational efficiency and convergence, we pruned the SNPs before running ADMIXTURE. For blue wildebeest, we removed sites with MAF lower than 0.05 or in high LD ($r^2 > 0.7$), which was implemented by "--maf 0.05 --indep-pairwise 1000 100 0.7" in PLINK. The input dataset was further thinned to 0.9 million SNPs by random selection. For black wildebeest, we filtered out sites with MAF lower than 0.05, missing call frequencies greater than 0.05 or in high LD ($r^2 > 0.8$), which was implemented by "--maf 0.05 --geno 0.05 --indep-pairwise 1000 100 0.8" in PLINK. For each K, we independently ran the ADMIXTURE analysis 200 times. The convergence criterion was defined as having the top 3 maximum likelihood runs within 5 log-likelihood units of each other.

## Population homogeneity

Population units of the blue wildebeest were determined according to clustering of the ADMIXTURE model with the highest K at which the model converged. We examined model fit by visualizing pairwise correlations of residuals with evalAdmix[30]. To investigate whether a bad model fit was caused by closely related samples within a population, we estimated the degree of relatedness based on identity-by-descent (IBD) coefficients for each of the populations with substantial non-zero correlations of residuals using PLINK v1.9. Relatedness inference was based on dataset1, where we further discarded sites with MAF < 0.05 and missingness > 5% within each population. We subsequently filtered out any related sample of second degree, and re-evaluated model fit for the ADMIXTURE analyses for the retained samples.

For the black wildebeest ADMIXTURE and evalAdmix indicated a diffuse population structure (Figs S7−S8). Therefore, we inferred its population units based on both geographical and genetic context. Individuals with more than one ancestry component in the ADMIXTURE model with K = 2 were excluded to form a more homogeneous population sample. To further assess homogeneity of the sampling sites of black wildebeest, we calculated D-statistics with individuals from the same location as H1 and H2, a neighboring Brindled population of blue wildebeest (B-Kalahari) as H3 and hartebeest as an outgroup. Samples consistently causing non-zero D-values were considered as outliers.

Further, we inspected patterns of LD decay for all populations that contain at least five samples. Briefly, for each of these populations, we calculated its LD in R using the *relate*[89] library and summarized the result in bins to generate an LD decay curve based on polymorphic sites in chromosome 2 (dataset1). Prior to the analysis, we thinned the data to 10% of the original sites using the "--thin 0.1" function in PLINK. To avoid potential bias due to varied sample sizes among populations, we downsampled all the tested populations to five individuals[90]. Pairwise LD was calculated using a window of 36,000 SNPs to reach a physical distance of 20 Mb, at which most decay curves plateaued. Taken together, the preceding analyses of population homogeneity identified nine homogenous populations in blue wildebeest and one population in black wildebeest (see more details in Supplementary Note 1).

## mtDNA analyses and phylogenetic tree

To construct mitochondrial sequences for the 131 wildebeest samples, we performed consensus calling based on reads aligned to the scaffold

"NC_020699.1", which represents the mitochondrial genome of blue wildebeest, using bcftools. Bases with quality scores lower than 30 in the constructed sequences were masked as "N" with seqtk (https://github.com/lh3/seqtk). We also downloaded mitochondrial genomes of hartebeest (*Alcelaphus buselaphus*; GenBank ID: NC_020676.1), topi (*Damaliscus lunatus*; NC_023543.1) and Gemsbok (*Oryx gazella*; NC_016422.1) from NCBI as outgroups. We then used MUSCLE v3.8.425 with default settings to conduct multiple alignment for all the mitochondrial genomes[91]. ModelTest-NG v0.1.7[92] was used to determine the substitution model, and HKY + I + G4 was chosen as the best-fitting model with a proportion of invariable sites of 0.6706 and a gamma distribution shape parameter of 0.651, which were used as priors in the phylogeny reconstruction. Phylogenetic analysis was performed using BEAST2 v2.7[93] based on the alignment of 16,713 sites. We used the Calibrated Yule Model constraining the tree with two node calibrations: 1) root calibration based on a previous estimate of time to most recent common ancestor (TMRCA) of the gemsbok, hartebeest and wildebeest drawn from a lognormal distribution with an expected mean value of 8.83 My and a standard deviation of 0.883[94]; and 2) a node calibration of TMRCA of hartebeest and wildebeest drawn from a lognormal distribution with an expected mean value of 3.23 My and a standard deviation of 0.323[94]. The analyses were run with a chain length of 5,000,000 steps, with a pre-burnin of 1000, and storing trees every 5000 steps. A maximum clade credibility tree was generated in TreeAnnotator with 30% burn-in, posterior probability limit of 0.7, and using the common ancestor heights. The convergence of the run was verified in Tracer v1.7.1[95] and all Effective Sample Sizes reached values above 100.

## Population differentiation

To quantify the extent of genetic differentiation among wildebeest populations, we calculated Hudson's estimator of genome-wide $F_{ST}$[96] based on dataset1. The analysis was performed using PLINK v2.0[97] with the following flag "--fst CATPHENO method=hudson".

## Estimation of effective migration surfaces

We conducted an Estimation of Effective Migration Surfaces (EEMS) analysis[43] to identify potential gene flow barriers among populations of the blue wildebeest. As an input to EEMS, we calculated the matrix of IBS distances between all pairwise blue wildebeest samples based on dataset1 using the "bed2diffs" script, as integrated in EEMS. We ran EEMS with the settings of 30 million steps of the chain including 15 million burn-in iterations and 600 demes. To ensure convergence, six independent Markov chain Monte Carlo (MCMC) chains were performed. The result of the EEMS was visualized using a custom R script based on the *reemsplot* package (http://www.github.com/dipetkov/eems).

## PSMC and PopsizeABC

We estimated effective population sizes back through time using PSMC[31]. We applied PSMC on all samples with a minimum average depth of 14 using dataset2. To scale the results for visualization, we used a generation time of 7.5 years and mutation rate of $1.45 \times 10^{-8}$ per generation following Table S16 in Chen (2019)[80].

Since PSMC has low resolutions for recent history, we applied PopSizeABC[32] on the populations with sufficient sample size ($n \geq 9$) to infer recent changes in population sizes. Based on dataset1, we used PopSizeABC to summarize two classes of statistics including the average LD and folded site frequency spectrum at different physical distances for each tested population. From this, we then ran PopSizeABC to estimate the effective population size through time in an approximate Bayesian computation (ABC) framework. We ran PopSizeABC on each population with 210,000 simulations and 100 regions of 2 Mb per simulation as per the recommended settings in Boitard (2016)[32]. A minimum MAF threshold of 0.1 was applied for the

estimation of the site frequency spectrum and a threshold of 0.2 for calculation of the LD.

## Heterozygosity and runs of homozygosity

We assessed genetic diversity of wildebeest based on levels of individual genome-wide heterozygosity. Individual heterozygosity was measured as the proportion of heterozygous loci per sample based on dataset2. We further inferred runs of homozygosity (ROH) using dataset3 with additional filters. We first masked all heterozygous genotypes with allelic balance (AB) below 0.25 or above 0.75. We next filtered for SNPs with a minimum MAF of 0.05, a maximum data missingness of 5% and a maximum ratio of observed heterozygotes of 50% within each subspecies. Detection of ROH was performed using the "--homozyg" function in PLINK v1.9 with default settings, except a maximum of 3 heterozygous SNPs and 20 missing calls within a scanning window. Validity of identified ROH was visually inspected by plotting SNP calls accompanied with the proportion of heterozygous sites and SNP density in sliding windows along each chromosome with a custom R script (Figs S22–S27). To investigate the hypothesized recent inbreeding in the black wildebeest that experienced a severe bottleneck, we made a rough estimate of the number of generations since the inbreeding took place based on the distribution of identified ROH. According to Howrigan (2011)[98] the lengths of autozygous segments, which are highly related to segments of ROH, should follow an exponential distribution with the expected length equal to $1/(2g)$ Morgan, where g is the number of generations since inbreeding. Assuming an average recombination rate of 1.275 cM/MB as estimated in cattle[99], we calculated the probabilities of a ROH being longer than 10 Mb (equivalent to 12.75 cM in wildebeest) with different numbers of generations. The probability of any one ROH being longer than 10 Mb is very low (0.674%) with 25 generations since the inbreeding took place.

## D-statistics, qpGraph, and outgroup f3

To infer the evolutionary relationships and past migration events among the inferred wildebeest populations, we utilized Dsuite[35] and ADMIXTOOLS2[36,100] based on dataset4. Both analyses were restricted to homogenous populations and based on the SNPs mapped to goat reference genome to mitigate effects of reference bias on estimation of population allele sharing. We calculated Patterson's *D* (also called ABBA-BABA) and the related *F*-branch statistics using the software package Dsuite to test for introgression between wildebeest populations. We included the two hartebeest samples as an outgroup. Calculations were limited to the groupings of populations that fit with a supplied neighbor-joining tree based on IBS distances using the Dtrios function.

To further investigate directionality and extent of gene flow, we carried out qpGraph analysis using the R package ADMIXTOOLS2. We ran qpGraph on a subset of homogeneous populations including the Etosha population as a representative of the Brindled wildebeest and all populations in the other subspecies of blue wildebeest to reduce model complexity. As input, we calculated allele frequencies and f2 statistics (f2s) for each population in a block of 4 Mb. We then ran the function "find_graphs" in qpGraph to explore admixture graphs with the assumed number of admixture events ranging from 0 to 5. We used a testing procedure where we first found the best scoring graph out of 400 candidate graphs for a given number of migrations and identified the graphs with the same number of migrations which are not significantly worse ($p > 0.05$). We also tested whether the best scoring graph for each number of migrations could be significantly rejected in favor of a graph with a higher number of migrations. The test score was calculated by optimizing a topology from a subset of the f-statistics and evaluating on the remaining. Test of significance was performed using a jackknife approach for each obtained graph, which was compared to the remaining graphs using a nominal *P*-value of

0.05. We discarded graphs with temporally implausible admixture events, where the first donor of an admixture event is also a descendant of the other donor. The best number of admixture events was determined by the best scoring graph that is significantly better than all graphs with a lower number of migrations.

Additionally, we calculated outgroup f3 statistics based on dataset4 using ADMIXTOOLS2[36] to investigate gene flow between the black wildebeest and blue wildebeest populations. With hartebeest as an outgroup, we estimated outgroup f3 in blocks of 5 Mb.

### D-statistic frequency spectrum

To specifically test presence and directionality of gene flow between species, we estimated the $D$ frequency spectrum ($D_{FS}$), which partitions $D$-statistics according to the frequencies of derived alleles[37]. The underlying model assumes occurrence of introgression, if not extremely archaic, to cause a peak at low-frequency derived alleles. We estimated the $D_{FS}$ using dataset4 with a pre-defined population tree composed of the Western white-bearded population as H1, Brindled population in Etosha as H2 and black wildebeest as H3. Hartebeest was included as an outgroup to polarize alleles for the ingroup populations.

### Local ancestry

As introgression analyses indicated ancient admixture from the black wildebeest into the Brindled wildebeest, we estimated distributions of local ancestries in the Brindled population in Etosha. We used the software package Loter[101] to perform local ancestry inference with imputed and phased SNPs (dataset1). The homogenous population of black wildebeest and West white-beared wildebeest were used as parental populations, representing the ancestry of the black and blue wildebeest, respectively, to reconstruct ancestry tracts in each B-Etosha sample. As control, we also inferred local ancestry in the Nyassa population with the same parental populations.

### Demographic modeling based on SFS

To date the interspecific admixture event, we modeled the demographic history of wildebeest populations using fastsimcoal v2.7.0.2[102]. To meet the requirement of fastsimcoal2 for non-recently admixed populations, we included three homogenous populations including the Brindled population in Etosha, Western white-bearded, and the black wildebeest in the demographic modeling. To prepare input, we estimated the folded site frequency spectrum (3d-SFS) using both variable and invariable sites mapped to blue wildebeest genome, where only sites without missing data were included, with a minimum of ten reads supporting all genotypes and a minimum of two reads supporting each allele of heterozygous genotypes. To explore the optimal demographic scenario, we considered three different models including 1) one-pulse gene flow event from the black wildebeest into B-Etosha with a fixed admixture proportion of 12% according to the qpGraph results, 2) a model of the same one-pulse gene flow event with admixture proportion being freely estimated, 3) a model of continuous gene flow from the black wildebeest into B-Etosha. We optimized each model using 100 independent runs with the following settings: 500,000 coalescent simulations per likelihood estimation (-n500000), 100 conditional maximization algorithm cycles (-L100), and a minimum of 100 observed SFS entry count considered for likelihood computation (-C100). We visually examined the model fit for the best likelihood run by plotting residuals for marginalized 1d and 2d-SFSs. As all three models yielded very similar likelihoods after optimization, we chose the simplest model (model 1) as the optimal model. To obtain confidence intervals for the parameters in model 1, we generated 50 bootstrapping 3d-SFSs by resampling bins of 0.5–1.5 million adjacent sites. For each bootstrap sample, we performed 50 independent runs of optimization with the same settings as above to ensure convergence. Based on the estimates from optimal runs, we

calculated the standard errors (SE) of each demographic parameter and define the 95% confidence interval as 1.96 × SE.

### Selection scan

To search for loci putatively under selection, we investigated the genomic landscapes between the blue wildebeest and black wildebeest. To depict the patterns of interspecific differentiation, we used the python package scikit-allele (https://github.com/cggh/scikit-allel) to estimate Hudson's $F_{ST}$ between the Brindled population in Etosha and the population of black wildebeest in non-overlapping windows of 100 kb based on dataset1. We further utilized the python package pixy[103] to inspect the local patterns of nucleotide diversity (π) in the two populations separately using the same window approach based on both variable and invariable sites, where any genotypes covered by less than ten reads and heterozygous genotypes supported by less than two reads for any of two alleles were masked. We further estimated patterns of LD decay in the region of exceedingly high $F_{ST}$. For comparison, we also estimated LD decay for the remaining loci on the same chromosome. Additionally, we examined the genes residing in the highly differentiated region based on the annotation of the blue wildebeest genome. To better understand the genomic features of this region, we performed pairwise alignment between the genomes of blue wildebeest and domestic goat using LASTZ[104].

## Data availability

Raw sequence data generated in this study have been deposited in the NCBI SRA database under accession code PRJNA1075443. Chromosome-level assemblies of the blue wildebeest (*C. taurinus*) and domestic goat (*Capra hircus*), which are available on the NCBI database with code GCA_006408615.1 and GCF_001704415.2, respectively, were also used in this study. Additional mitogenome sequences available in GenBank were utilized for mitochondrial analyses (*Alcelaphus buselaphus*: NC_020676.1, *Damaliscus lunatus*: NC_023543.1) and *Oryx gazella*: NC_016422.1). Source data are provided as a Source Data file. Source data are provided with this paper.

## Code availability

All the scripts used to analyze the data are available at https://github.com/popgenDK/seqAfrica_wildebeest with the https://doi.org/10.5281/zenodo.10679260 for version v1.0.0.

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

## Acknowledgements

We thank Amal Al-Chaer for her help with DNA extractions. We are indebted to Peter Arctander, who organized the collection of samples in the 1980s and 1990s. I.M., X.L. and R.F.B. are supported by a Villum Young Investigator grant (VIL19114) awarded to I.M. A.A., L.L., R.F.B. and Z.L. are funded by the Novo Nordisk Foundation (NNF20OC0061343). G.G.E. and R.H. are supported by a Danmarks Frie Forskningsfond Sapere Aude research grant (DFF8049-00098B), R.H. is further supported by a Carlsberg Young Researcher grant (CF21-0497). A.A. and M.S.R. are supported by the Independent Research Fund Denmark (grant number: 8021-00360B). A.B.O. is supported by a Carlsberg Foundation Reintegration Fellowship (CF19-0427). J.O.O. is supported by a grant from the German Research Foundation (DFG; grant number: 257734638). R.R.F is supported by the Villum Fonden (grant no 25925 for the Center for Global Mountain Biodiversity). R.R.P. is supported under a grant (2020.08608.BD) by the Portuguese Foundation for Science and Technology (FCT).

## Author contributions

X.L., H.R.S, A.A., I.M. and R.H. conceived the study; A.A., I.M. and R.H. supervised the study; X.L., L.L., L.D.B., K.H., L.Q., G.G.-E., M.S.R., M.S., P.P., R.F.B., Z.L., R.P., X.W., J.K., A.B.-O., J.M., C.S. and R.R.d.F analyzed the data; C.M., V.M., M.P.H. and T.P.L.S. provided tissue, sequencing data and crucial context for subset of the samples; X.L., M.-H.S.S, L.L., A.A., I.M. and R.H. drafted the manuscript with input from all authors; L.L., X.L. and L.D.B. created the figures; X.L., L.L., J.O.O., H.R.S, A.A., I.M. and R.H. revised the paper with input from other authors; all authors approved the paper for submission.

## Competing interests

The authors declare no competing interests.

## Additional information

[1]Department of Biology, University of Copenhagen, Copenhagen, Denmark. [2]Novo Nordisk Foundation Center for Basic Metabolic Research, University of Copenhagen, Copenhagen, Denmark. [3]USDA, ARS, U.S. Meat Animal Research Center (USMARC), Clay Center, NE, USA. [4]CIIMAR—Interdisciplinary Centre of Marine and Environmental Research—University of Porto, Porto, Portugal. [5]Section for Biodiversity, Globe Institute, University of Copenhagen, Copenhagen, Denmark. [6]Copenhagen Research Centre for Mental Health, Copenhagen University Hospital, Copenhagen, Denmark. [7]Biostatistics Unit, Institute of Crop Science, University of Hohenheim, Stuttgart, Germany. [8]Department of Zoology, Entomology and Fisheries Sciences, Makerere University, P. O. Box 7062 Kampala, Uganda. [9]Department of Environmental Management, Makerere University, PO Box 7062 Kampala, Uganda. [10]These authors contributed equally: Xiaodong Liu, Long Lin, Mikkel-Holger S. Sinding. [11]These authors jointly supervised this work: Anders Albrechtsen, Ida Moltke, Rasmus Heller. ✉e-mail: aalbrechtsen@bio.ku.dk; ida@bio.ku.dk; rheller@bio.ku.dk

