## [Peer Review File · Nature Communications]

Introgression and disruption of migration routes have shaped the genetic integrity of wildebeest populationsThis manuscript has been previously reviewed at another journal that is not operating a transparent peer review scheme. This document only contains reviewer comments and rebuttal letters for versions considered at *Nature Communications*.

REVIEWERS' COMMENTS

Reviewer #2 (Remarks to the Author):

The authors have addressed all of my concerns from the first round of edits in a thorough manner. I believe the current version of the manuscript is much improved and recommend acceptance.

Minor edit

Line 247-248: C-Selous and N-Luangwa in the text, but N-Selous and C-Luangwa in Figure 1A, please check which is correct.

Reviewer #3 (Remarks to the Author):

The authors have done a thorough job in responding to my comments on a previous version of this manuscript submitted to Nature Ecology & Ecology and I commend them for their efforts.

In particular, queries relating to sample quality and batch effects have been addressed through visualisation of sequencing depth versus heterozygosity.

The authors have attempted to deal with confusion surrounding population and sub-species names however I expect that these will still require effort to follow, since there are a many different wildebeest populations that are being compared here that won't be familiar to the average reader.

The authors have also carried out a reanalysis of genome-wide heterozygosity based on genomic regions outside of ROH, of which the results and interpretation make intuitive sense. Together with greater in-text clarification, this addresses my previous concerns relating to this topic. However, while I appreciate the correlation plots added to the supplementary material (Figure S12), because the scales on each x axis are standardised, it is impossible to see the actual range in the values.

I would also like to thank the authors for their explanation around the estimation of generation time based on IBD tracts. This section is now much clearer.

Finally, I appreciate the authors efforts in re-framing some of the text to focus more on the genetics of migratory behaviour in general. While the authors do this well, it is still not entirely clear to me how the work addresses a broader question with relevance to related areas and the manuscript is ultimately still largely focussed on wildebeest populations.

RESPONSE TO REVIEWERS' COMMENTS

Reviewer #2 (Remarks to the Author):

The authors have addressed all of my concerns from the first round of edits in a thorough manner. I believe the current version of the manuscript is much improved and recommend acceptance.

Minor edit

Line 247-248: C-Selous and N-Luangwa in the text, but N-Selous and C-Luangwa in Figure 1A, please check which is correct.

Responses: Thanks. We have corrected it.

Reviewer #3 (Remarks to the Author):

The authors have done a thorough job in responding to my comments on a previous version of this manuscript submitted to Nature Ecology & Ecology and I commend them for their efforts.

In particular, queries relating to sample quality and batch effects have been addressed through visualisation of sequencing depth versus heterozygosity.

The authors have attempted to deal with confusion surrounding population and sub-species names however I expect that these will still require effort to follow, since there are a many different wildebeest populations that are being compared here that won't be familiar to the average reader.

Responses: To help readers follow subspecies and population names, we have added the following sentence at the beginning of the Results section.

“These populations include B-Etosa, B-Kafue and B-Kalahari from Brindled, C-Luangwa from Cookson, N-Selous from Nyassa, E-Amboseli, E-Monduli and E-Nairobi from Eastern white-bearded, and W-Serengeti from Western white-bearded.”

Furthermore, when we refer to a population for the first time at each section of Results, we spell out the subspecies (name) to which it belongs. We think that these should help resolve confusion around population and subspecies names.

The authors have also carried out a reanalysis of genome-wide heterozygosity based on genomic regions outside of ROH, of which the results and interpretation make intuitive sense. Together with greater in-text clarification, this addresses my previous concerns relating to this topic. However, while I appreciate the correlation plots added to the supplementary material (Figure

S12), because the scales on each x axis are standardised, it is impossible to see the actual range in the values.

I would also like to thank the authors for their explanation around the estimation of generation time based on IBD tracts. This section is now much clearer.

Finally, I appreciate the authors efforts in re-framing some of the text to focus more on the genetics of migratory behaviour in general. While the authors do this well, it is still not entirely clear to me how the work addresses a broader question with relevance to related areas and the manuscript is ultimately still largely focussed on wildebeest populations.

Responses: We thank the reviewer for his/her appreciation of our efforts to re-frame the text to make it more broadly relevant. In addition to elucidating evolutionary histories of the wildebeest populations, our manuscript also provides new and general insight into the population genetic effects of anthropogenic activities on highly migratory ungulates, and in the previous revision we made substantial edits to convey this more general conclusion. In addition, we have now added the following sentences in Discussion in response to the reviewer comment.

“This finding raises concerns that the disruption of migration routes in naturally nomadic species leads to genetic erosion and long-term loss of evolutionary potential, in addition to the well-known short-term ecological impact.”